# Atomically Thin Amorphous Indium–Oxide Semiconductor Film Developed Using a Solution Process for High-Performance Oxide Transistors

**DOI:** 10.3390/nano13182568

**Published:** 2023-09-16

**Authors:** Jun-Hyeong Park, Won Park, Jeong-Hyeon Na, Jinuk Lee, Jun-Su Eun, Junhao Feng, Do-Kyung Kim, Jin-Hyuk Bae

**Affiliations:** School of Electronic and Electrical Engineering, Kyungpook National University, Daegu 41566, Republic of Korea; jeef1234@knu.ac.kr (J.-H.P.);

**Keywords:** amorphous oxide semiconductor, thin-film transistors, ultrathin channel, high performance, solution process

## Abstract

High-performance oxide transistors have recently attracted significant attention for use in various electronic applications, such as displays, sensors, and back-end-of-line transistors. In this study, we demonstrate atomically thin indium–oxide (InO*_x_*) semiconductors using a solution process for high-performance thin-film transistors (TFTs). To achieve superior field-effect mobility and switching characteristics in TFTs, the bandgap and thickness of the InO*_x_* were tuned by controlling the InO*_x_* solution molarity. As a result, a high field-effect mobility and on/off-current ratio of 13.95 cm^2^ V^−1^ s^−1^ and 1.42 × 10^10^, respectively, were achieved using 3.12-nanometer-thick InO*_x_*. Our results showed that the charge transport of optimized InO*_x_* with a thickness of 3.12 nm is dominated by percolation conduction due to its low surface roughness and appropriate carrier concentration. Furthermore, the atomically thin InO*_x_* TFTs showed superior positive and negative gate bias stress stabilities, which are important in electronic applications. The proposed oxide TFTs could provide an effective means of the fabrication of scalable, high-throughput, and high-performance transistors for next-generation electronic applications.

## 1. Introduction

Metal–oxide thin-film transistors (TFTs) were introduced for use in displays, sensors, and back-end-of-line (BEOL) transistors due to their various advantages, such as low processing temperature, high mobility, low subthreshold swing (SS), optical transparency, and high mechanical flexibility [1,2,3]. Oxide semiconductors in the solution process have attracted a lot of attention because they enable low-cost and high-throughput manufacturing, although sputtered amorphous indium–gallium–zinc–oxide semiconductors are more widely used in industry due to their high uniformity, good electrical performance, and bias stability [4,5,6]. Unfortunately, it is difficult to fabricate high-performance amorphous oxide TFTs using the solution process. Although multicomponent metal oxides are usable to form the amorphous phase, which leads to high uniformity and yield in mass production, a high annealing temperature of over 300 °C is generally required to promote metal–oxygen–metal condensation and densification in solution-processed multicomponent oxide TFTs [7,8]. In addition, semiconductors with relatively low electron mobility are produced via this process due to the poor quality of the solution-processed films compared to vacuum-processed films.

To overcome these issues, various attractive methods have been proposed in terms of the fabrication process, synthesis, and device configuration. For example, combustion synthesis and photochemical activation were introduced for low-temperature fabrications [5,9,10]. Heterostructure semiconductors and various metal doping strategies were proposed to create high-performance transistors [11,12]. With similar goals in mind, various oxide materials have been suggested. Indium oxide (InO*_x_*) is one of the most desirable semiconducting materials due to its high intrinsic electron mobility and relatively low activation temperature (200–300 °C) [13,14]. Despite these advantages, its poor switching characteristics with high off-current and the polycrystalline structure are critical obstacles to the practical use of InO*_x_* TFTs. Controlling the semiconductor thickness at the atomic scale allows the effective tuning of the bandgap and carrier concentration of the semiconductor, thereby enabling the achievements in terms of reasonable switching characteristics, mobility, and bias stability in transistors [15]. Atomic-layer-deposited high-quality InO*_x_* of several nanometers in thickness have been reported for BEOL-compatible transistors [3,16]. However, the formation of atomically thin amorphous InO*_x_* semiconductors through a low-temperature solution process and the high-performance amorphous oxide transistor based on it have not yet been thoroughly studied.

In this study, we address these challenges by developing a solution-deposited ultrathin amorphous InO*_x_* semiconductor. By controlling the molarity of the solution, InO*_x_* semiconductors were deposited at various thicknesses in the range 1.95–5.51 nm. Their crystallinities, bandgaps, atomic bonding states, and electrical characteristics are also presented. Despite being fabricated in a solution processed at a low temperature of 300 °C, InO*_x_* TFTs with optimal thicknesses of about 3 nm exhibited high field-effect mobility (*μ*_FE_) and an on/off-current ratio (*I*_on/off_) of 13.95 cm^2^ V^−1^ s^−1^ and 1.42 × 10^10^, respectively. We show that the charge transport of the roughly 3-nanometer-thick low-dimensional InO*_x_* is dominated by percolation conduction due to low surface roughness and appropriate carrier concentration. The bias stability values of atomically thin InO*_x_*-based TFTs are also investigated. This achievement shows the possibility of overcoming the limitations of high-mobility semiconductor materials and solution processes.

## 2. Materials and Methods

### 2.1. Preparation of Oxide Precursor Solutions

InO*_x_* precursor solutions were prepared for the fabrication of oxide semiconductors. InO*_x_* solutions were prepared by dissolving In(NO_3_)_3_·*x*H_2_O (Sigma-Aldrich, St. Louis, MO, USA) 2-methoxyethanol at various molarities of 0.05, 0.07, 0.09, and 0.11. All solutions were stirred at a room temperature of 25 °C for 30 min to obtain clear and homogeneous solutions.

### 2.2. Fabrication of TFTs

Si/SiO_2_ wafers were cleaned using sonication for 10 min each in acetone, isopropyl alcohol, and deionized (DI) water. The residual moisture on the cleaned substrates was removed via drying with N_2_ and annealing at 300 °C for 30 min. We previously reported the use of a water etchant-based photopatterning method to deposit the patterned low-defect oxide thin films, which is compatible with the solution process [17]. In this study, water etchant-based photopatterning was used to deposit the high-quality InO*_x_* semiconductors. Figure 1 displays a schematic of the process of the fabrication of oxide TFTs. InO*_x_* solution was spin coated at 3000 rpm for 20 s on a cleaned wafer (Step 1). The InO*_x_*-deposited wafer was then soft baked at 100 °C for 30 s (Step 2). Afterward, a fine metal mask was covered for selective exposure to ultraviolet (UV) light for 60–120 s in an N_2_ environment (Step 3). UV irradiation (25 mW cm^−2^) was conducted via a low-pressure mercury lamp with two main wavelengths, namely 253.7 (90%) and 184.9 nm (10%). Consecutively, the wafer was developed in a DI water etchant for 1 min (Step 4). Finally, clearly patterned InO*_x_* was subsequentially annealed at 100 °C for 10 min and 300 °C for 2 h on a hot plate in air (Step 5). In this series of processes, UV photons induced the photochemical cleavage of alkoxy groups and decomposition nitrate ligands; therefore, the UV-irradiated parts remained in the etching process. Also, water molecules were sufficiently supplied to the InO*_x_* semiconductor during the etching process, thereby enabling effective hydrolysis and condensation reactions at the annealing stage, as shown in Appendix A. As a result, the reactions promote the densification and defect reduction in InO*_x_*, which are advantageous for high-performance oxide transistors [17]. The final InO*_x_* thicknesses obtained were in the range 1.95–5.51 nm depending on the solution molarity. The 50-nanometer-thick Al was deposited using thermal evaporation for source and drain electrodes. The channel length (L) and width (W) were 100 and 1000 μm, respectively. Here, the molarities of InO*_x_* were 0.05, 0.07, 0.09, and 0.11, and the corresponding InO*_x_* thin films and TFTs were named InO*_x_*-5, InO*_x_*-7, InO*_x_*-9, and InO*_x_*-11, respectively.

### 2.3. Analysis of Thin Films and Devices

Transmission electron microscopy (TEM; FEI Company, Hillsboro, OR, USA, Titan G2 ChemiSTEM Cs Probe) was used to observe the film’s thickness, crystallinity, and elemental mapping. The surface morphologies of the atomically thin InO*_x_* were observed using non-contact-mode atomic force microscopy (AFM; Park Systems, NX20, Suwon, Republic of Korea). The chemical compositions of the InO*_x_* were analyzed using X-ray photoelectron spectroscopy (XPS; ThermoFisher Scientific, NEXSA, Waltham, MA, USA) with an Al Kα (1486.6 eV) X-ray source. UV–Visible (UV–Vis) spectrophotometry (Perkin Elmer, LAMBDA 265, Waltham, MA, USA) was measured to understand the effects of thickness at the atomic scale on the optical bandgap of InO*_x_*. Grazing incidence X-ray diffraction (GIXRD) (Bruker, D8 Discover, Billerica, MA, USA) tests were carried out using Cu Kα radiation and a small incident angle of 0.5° to investigate the crystalline phase characteristics of InO*_x_*. The electrical characteristics and bias stabilities of the InO*_x_* TFTs were measured using a probe station (MS Tech, model 4000, Seoul, Republic of Korea) with a precision LCR meter (Agilent E4980A, Santa Clara, CA, USA) and a semiconductor parameter analyzer (Keithley 2636B, Cleveland, OH, USA). All devices were measured at room temperature in a dark environment.

## 3. Results and Discussion

The physical characteristics of atomically thin InO_x_ semiconductors were investigated. The InO_x_ semiconductors were deposited using solutions with low molarities, namely 0.05, 0.07, 0.09, and 0.11 M, to obtain films with sub-10-nanometer thicknesses, as depicted in Figure 1. All InO_x_ thin films were deposited on Si/SiO_2_ wafers to enable physical analysis, and film thickness was controlled by fine-tuning the solution molarity, as shown in Figure 2a. As shown in the TEM images of InO_x_-5, InO_x_-7, InO_x_-9, and InO_x_-11 in Figure 2b, the InO_x_ thickness gradually decreases as the molarity decreases. Interestingly, while binary oxides usually have a polycrystalline or nanocrystalline phase, all InO_x_ films in this study exhibit an amorphous phase, as confirmed in selected area electron diffraction (SAED) patterns, which is favored by industry because it enables the realization of large-area displays and high uniformity and yield (inset of Figure 2b) [18]. In particular, InO*_x_* typically crystallizes in a cubic structure even at low annealing temperatures [19]. This unique result is attributed to the supply of water molecules to InO*_x_* at the initial stage of film formation during the etching step (Appendix A) [17,20]. The amorphous phase is also confirmed via GIXRD patterns, which agree with the SAED patterns (Appendix A). Element mapping was carried out for O, In, and Si. Element mapping showed that the InO*_x_* thickness gradually decreased as molarity increased (Appendix A). AFM images (Figure 2c, top) and the corresponding height profiles (Figure 2c, bottom) of InO*_x_*-5, InO*_x_*-7, InO*_x_*-9, and InO*_x_*-11 are shown in Figure 2c to confirm the surface morphology of ultrathin InO*_x_* at various solution molarities. The yellow dashed line in the AFM images indicates where the height profiles were measured. All samples show smooth surfaces because the films were deposited via spin coating, which is a desirable method used to form thin and uniform films. The correlations between molarity, thickness, and roughness were demonstrated. Figure 2d,e presents the correlations between molarity and thickness and thickness and roughness, respectively. The average thicknesses of InO*_x_*-5, InO*_x_*-7, InO*_x_*-9, and InO*_x_*-11 were 1.95, 3.12, 4.40, and 5.51 nm, respectively. As expected, the thickness linearly increased with the increase in the molarity because molarity indicated the concentration of indium in the solution (Figure 2d). Although the spin coating speed was also one of the methods used to control semiconductor thickness, the controllability of the thickness via this method was poor because the coating speed and thickness did not have linearity. Therefore, the molarity was controlled in this study because it has linearity with thickness. We noted that due to the low viscosity of solution and high spin speed, there was a negligible difference in the thickness of the center and edge regions within the substrate, showing high thickness uniformity. The R-squared (R^2^) was extracted to examine how well an estimated linear model fit a given dataset. An R^2^ of 1.00 was obtained for the molarity–thickness plots, implying a very high correlation. The average root-mean-square (RMS) surface roughnesses of InO*_x_*-5, InO*_x_*-7, InO*_x_*-9, and InO*_x_*-11 were 0.122, 0.123, 0.125, and 0.132 nm, respectively. The RMS roughness was linearly increased by increasing the semiconductor thickness at the atomic scale and showing a high R^2^ of 0.98 in the roughness–thickness plot (Figure 2e).

Optical bandgaps (E*_g_*) of InO*_x_* with various thicknesses were investigated using a UV–Vis spectrophotometer. The E*_g_* was extracted via the extrapolated linear fit of (αhν)^2^ versus the photon energy, as shown in Figure 3a. The extracted E*_g_* of InO*_x_*-5, InO*_x_*-7, InO*_x_*-9, and InO*_x_*-11 were 3.92, 3.92, 3.90, and 3.88 eV, respectively. The E*_g_* of InO*_x_* tended to increase as the thickness became thinner, which was mainly explained by the quantum confinement effect (Figure 3b) [21,22]. Changes in the chemical bonding states of InO*_x_* with various solution molarities were also investigated using XPS analysis. Figure 3c shows the O1s spectra of InO*_x_*-5, InO*_x_*-7, InO*_x_*-9, and InO*_x_*-11. The oxygen bonding states were analyzed via deconvolution into three Gaussian peaks corresponding to the oxygen species of the metal–oxide (M–O), oxygen vacancy (*V*_o_), and metal–hydroxide (M–OH). The three main Gaussian curves of M–O, *V*_o_, and M–OH were centered at approximately 530.3 ± 0.2, 531.5 ± 0.2, and 532.5 ± 0.2 eV, respectively [23,24,25,26]. We noted that the curve at about 532.5 eV could be interpreted as Si–O [27,28]. Si atoms of SiO_2_ usually diffused into the oxide semiconductor during the annealing process in metal–oxide TFT fabrications, meaning that Si–O bonding might exist at the interface between the SiO_2_ dielectric and InO*_x_* semiconductor [28]. Here, the ratio of Si–O at the InO*_x_* surface gradually increases as the thickness of InO*_x_* decreases because InO*_x_* thin films are atomically thin with a sub-5-nanometer thickness. In addition, since all processes, except for solution molarity, were identically performed and simultaneously fabricated in the same environment, it is reasonable to interpret the peak at 523.5 eV as Si–O rather than M–OH.

Atomically thin amorphous InO*_x_* TFTs with various solution molarities were fabricated to demonstrate their electrical characteristics. Figure 4a shows the transfer characteristics of InO*_x_*-5, InO*_x_*-7, InO*_x_*-9, and InO*_x_*-11. To investigate the transfer characteristics, the gate voltage (*V*_G_) was varied from −20 to 30 V, while the drain voltage (*V*_D_) was fixed at 30 V. The solid and dotted lines in Figure 4a show the drain (*I*_D_) and gate currents (*I*_G_), respectively. The threshold voltage (*V*_T_) was calculated by fitting a straight line to the plot of the square root of *I*_D_ versus *V*_G_ in the saturation region (Appendix A). As the thickness of InO*_x_* increased, the *V*_T_ negatively shifted and *μ*_FE_ increased. The large negative shift of *V*_T_ with increasing molarity originated from an increase in the total number of carriers in the semiconductor. Although the electron concentration was the same, the total number of electrons increased as the semiconductor thickness increased, and channels were formed relatively easily as *V*_G_ increased. Although InO*_x_*-5 had a low off-current level of 10^−12^ A and a positive *V*_T_, which is desirable in circuit design, a very low *μ*_FE_ of InO*_x_*-5 might be insufficient to operate the electronic devices [29,30]. On the other hand, InO*_x_*-9 and InO*_x_*-11 had high on-currents, but their inferior switching characteristics were not suitable for switching transistors in applications such as displays. On the other hand, InO*_x_*-7 had a high *μ*_FE_ and *I*_on/off_ of 13.95 cm^2^ V^−1^ s^−1^ and 1.42 × 10^10^, respectively, meaning that it was considered to be an optimized device with a high electrical performance. The slight difference in *I*_G_ for each device might have originated from the subtle differences in the thermally grown SiO_2_ film quality rather than the InO*_x_* layer characteristics or fabrication process.

To deduce the charge transport mechanism in the atomically thin InO*_x_* TFTs, *μ*_FE_ in relation to *V*_G_ is shown in Figure 4b. The gradual decrease in *μ*_FE_ with increasing *V*_G_ in InO*_x_*-5 suggested that surface roughness scattering dominated in ultrathin semiconductor-based TFTs [21]. Thus, the fluctuation in potential energy due to surface roughness has a high effect on long-range electron transport. In InO*_x_*-7, *μ*_FE_ gradually increased as *V*_G_ increased, especially showing a power law behavior [17,31,32]. At high *V*_G_, the exponent *γ* extracted through power law fitting was 0.34, indicating that electron transport was based on percolation conduction. An increase in *V*_G_ led to an increase in electron concentration in the channel, which meant that the Fermi level was closer to the conduction band. As the Fermi level increased, the interference of electronic conduction of the potential barrier became insignificant. As the thickness increased, the bandgap became narrower, and the number of carriers in the semiconductor increased. Therefore, when the same *V*_G_ was applied, the electron concentration of the channel was relatively high in oxide TFTs, having a thick semiconductor film compared to that of the oxide TFTs with thin semiconductor films. Consequently, the gap between the Fermi level and conduction band minimum was closer in thick semiconductor. Therefore, the potential barrier formed above the conduction band minimum appeared to be relatively low from the perspective of electron [32]. Therefore, as the thickness of InO*_x_* increased, the percolation effect in InO*_x_* decreased, and the band-like transport became dominant. The extracted γ value in the power law model of InO*_x_*-7 was 0.34, and it indicated that the electron transport was governed by the percolation conduction because it was generally understood that the percolation conduction is dominant when γ was close to 0.1 [31]. In several of the studies reported in the literature related to percolation conduction, the γ values were in the range of 0.12 to 0.34, which is comparable to this study [33,34,35,36]. Meanwhile, a negligible change in *μ*_FE_ occurred with an increase in *V*_G_ in InO*_x_*-9 and InO*_x_*-11. This result implied that controlling the semiconductor thickness at the atomic scale allowed the effective tuning of electrical characteristics. In addition, it was confirmed that InO*_x_* TFTs were saturated in the *V*_D_ ≥ *V*_G_ − *V*_T_ region by output characteristics, as shown in Figure 4c. In addition, the device-to-device uniformity of optimized TFTs InO*_x_*-7 was investigated by measuring 30 devices through the fabrication of three InO*_x_*-7 samples (10 devices per sample), as shown in Appendix A. Appendix A shows the schematic illustration of fabricated TFT arrays and measurement points used for statistical evaluation. As shown in Appendix A, considering that the devices were fabricated through a solution process, the reasonable high device-to-device uniformity was obtained, although the semiconductor thickness was at the atomic scale [17,37]. This result might be attributed to the fact that the InO*_x_* used in this study had high film quality with few physical and chemical defects due to the densification and condensation effects incurred via water etchant [17].

The electrical performances of atomically thin InO*_x_* TFTs used in this work were benchmarked with recently reported high-performance oxide TFTs, as shown in Figure 5. All of the data presented here were derived from the solution-processed oxide TFTs with SiO_2_ dielectrics, since mobility overestimation via high-*k* dielectrics could affect quantitative comparison. The semiconductor materials, processing temperature, and electrical characteristics presented are in Figure 5 and summarized in Appendix A. We noted that the operating conditions, such as biased range or value, are not exactly the same for each study, but almost all of them were driven under the similar conditions that could be acceptable in quantitative comparison. TFTs fabricated in this study showed high *μ*_FE_ at a relatively low processing temperature and positive *V*_T_ (Figure 5a,b). From an industrial perspective, the processing temperature of 300 °C enabled the utilization of flexible polymer substrates and minimized the process risks. Furthermore, it allowed the back-end-of-line fabrication. Thus, it could be considered to be a sufficiently low processing temperature in the display and semiconductor industries [38]. Furthermore, a positive *V*_T_ was preferred to a negative *V*_T_ because it was advantageous in circuit design and power consumption [29,30]. As shown in Figure 5c, oxide TFTs used in this work had a superior *I*_on/off_ of about 10^10^ using the high *μ*_FE_. High *I*_on/off_ contributed to the decrease in the leakage current when the transistor was in off state, and the high *μ*_FE_ allowed the lowering of the operating voltage to achieve the specific current required to drive the circuit or emit light. Therefore, the high *I*_on/off_ and *μ*_FE_ enabled an improvement in power efficiency. These comparisons imply that the proposed atomically thin amorphous InO*_x_* TFTs had considerable suitability as a candidate for use in a unit device in next-generation electronic device applications.

Bias stability is one of the most important factors in the commercial use of electronic devices. In this regard, the bias stress test was carried out to demonstrate the bias stabilities of the optimized atomically thin amorphous oxide TFT InO*_x_*-7. To demonstrate the positive bias stress (PBS)-induced instability of the InO*_x_*-7, *V*_G_s of 10 and 30 V were applied for 3000 s at room temperature, as shown in Figure 6a,b, respectively. A *V*_D_ of 0 V was applied during PBS tests. After stress, transfer characteristics were measured at a *V*_D_ of 30 V. Compared to PBS at a *V*_G_ of 10 V, the PBS at a *V*_G_ of 30 V experienced a more severe transfer curve shift due to strong gate bias stress. To understand the PBS-induced instability, variations in *V*_T_, SS, and *µ*_FE_ with stress time were extracted (Figure 6c). Regardless of the applied *V*_G_ during PBS, *V*_T_ shifted positively, while SS did not noticeably change as the stress time increased. In addition, *µ*_FE_ gradually decreased with the positive shift in *V*_T_. This result could be interpreted as meaning that the interface traps were not additionally created via bias stress, and the *V*_T_ shift originated from the electron trapping at the interface and electron injection into the gate dielectric (Figure 6d) [39,40]. For NBS tests, *V*_G_s of −10 and −30 V were applied for 3000 s at room temperature, as shown in Figure 6e,f, respectively. The *V*_D_ of 0 V was applied during the NBS test, and transfer characteristics were measured at *V*_D_ of 30 V, i.e., the same as the PBS tests. A negative shift in the *V*_T_ of InO*_x_*-7 was observed using the stressed *V*_G_s of −10 and −30 V. Interestingly, the change in SS was negligible when the stress *V*_G_ was −10 V, while it showed a gradual increase with the increasing stress time when the stress *V*_G_ was −30 V (Figure 6g). These different trends imply that when NBS at a *V*_G_ of −10 or −30 V was applied to InO*_x_*-7, the devices underwent degradation via different mechanisms. The fact that SS did not increase via NBS at a *V*_G_ of −10 V suggests that the *V*_T_ shift occurred due to hole trapping and injection rather than the degradation due to defect creation (Mechanism (1) in Figure 6h) [39]. On the other hand, the increase in SS via NBS at a *V*_G_ of −30 V indicated the formation of acceptor-like defects during NBS. In general, defects at deep levels, such as *V*_o_, in oxide semiconductors are ionized via illumination stress [39,41]. However, the Fermi level approached the valence band maximum via band bending due to the strong negative gate bias; the formation energy of *V*_o_^2+^ became lower than that of *V*_o_ [42]. Two electrons released via the transition from *V*_o_ to *V*_o_^2+^ induced a negative shift in *V*_T_. Meanwhile, the *V*_o_^2+^ migrated to the gate dielectric–semiconductor interface via the negative gate bias, thereby acting as an acceptor-like trap (Mechanism (2) in Figure 6h). Therefore, it was reasonable to estimate that the degradation mechanisms caused by NBS with *V*_G_ of −10 and −30 V were caused by the hole trapping and injection and *V*_o_ ionization, respectively, in solution-processed atomically thin amorphous InO*_x_* TFTs.

## 4. Conclusions

In this study, we demonstrated high-performance solution-processed oxide TFTs with atomically thin InO*_x_* semiconductors. Based on the linearity between solution molarity and film thickness, the thickness of InO*_x_* was precisely controlled by adjusting the molarity of the solution. As a result, ultrathin and uniform layers of 1.95–5.51 nm were deposited through the solution process. The bandgap of InO*_x_* increased as the thickness decreased due to the quantum confinement effect, and, thus, high *μ*_FE_ and *I*_on/off_ of 13.95 cm^2^ V^−1^ s^−1^ and 1.42 × 10^10^, respectively, were obtained in InO*_x_* TFTs at an optimized InO*_x_* thickness of 3.12 nm. Furthermore, atomically thin InO*_x_* TFTs showed superior positive and negative gate bias stress stabilities. Their degradation mechanisms were also investigated by analyzing the change in the electrical parameters via gate bias stress. The proposed atomically thin InO*_x_* TFTs are expected to contribute to the realization of high-performance printed transistors for use in next-generation electronic applications.

## Figures and Tables

**Figure 1 nanomaterials-13-02568-f001:**
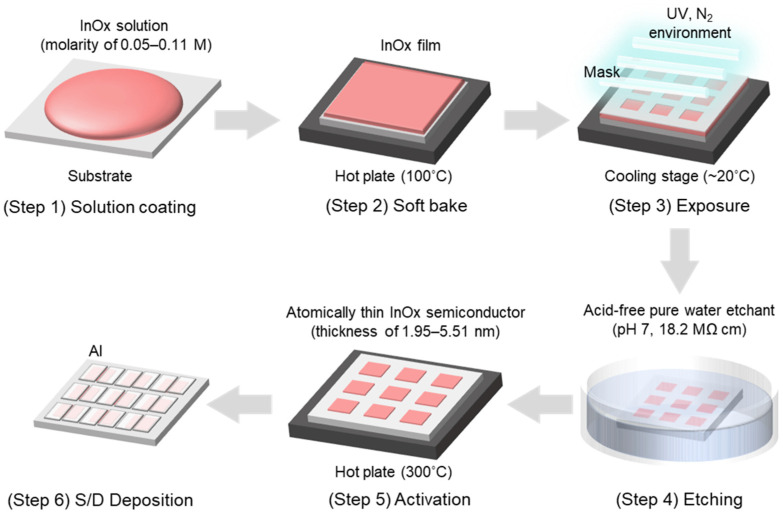
Schematic of the process of fabrication used for atomically thin InO*_x_* TFTs.

**Figure 2 nanomaterials-13-02568-f002:**
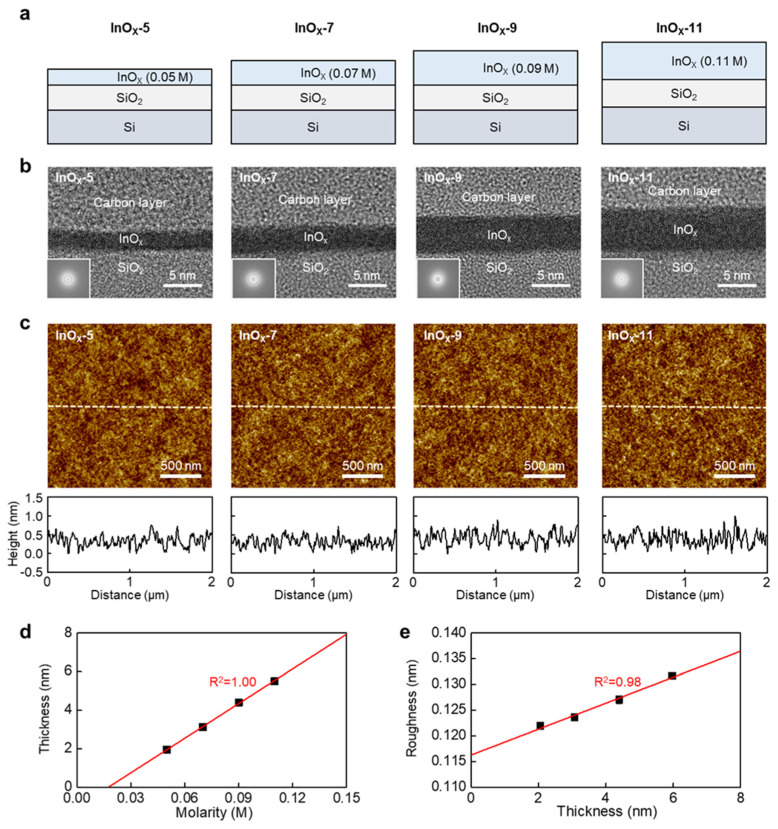
(**a**) Structures, (**b**) TEM images and corresponding SAED patterns (inset), and (**c**) AFM images and corresponding height profiles of InO*_x_* thin films at various molarities. (**d**) InO*_x_* film thickness versus molarity. (**e**) RMS roughness of InO*_x_* film versus InO*_x_* film thickness.

**Figure 3 nanomaterials-13-02568-f003:**
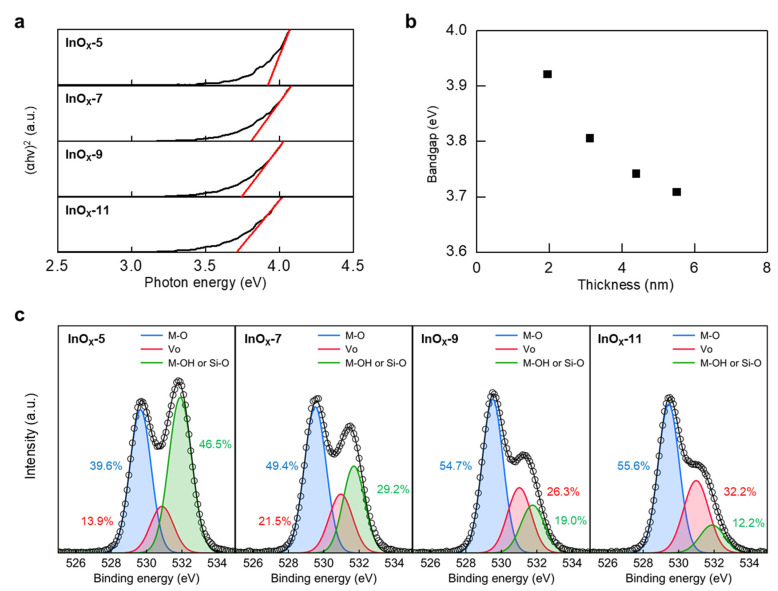
(**a**) (αhν)^2^ versus photon energy, (**b**) optical bandgap versus thickness, and (**c**) XPS O 1s spectra of InO*_x_* films with various solution molarities.

**Figure 4 nanomaterials-13-02568-f004:**
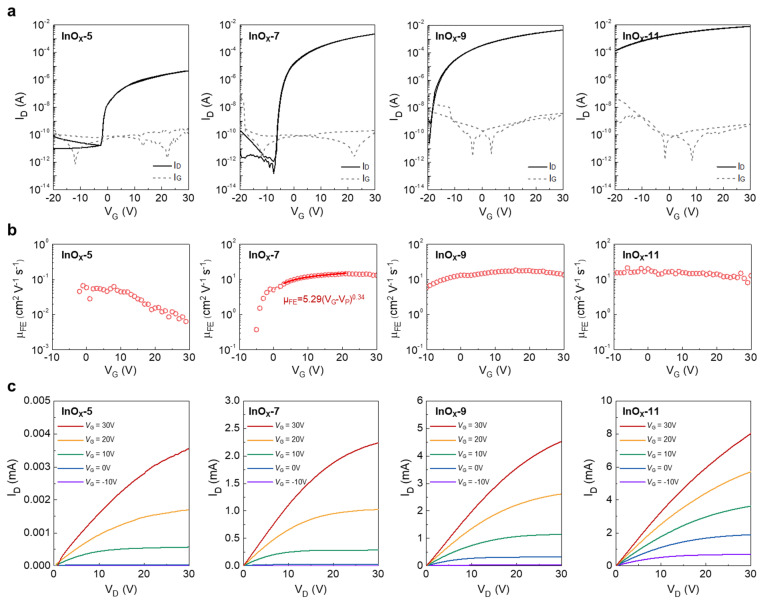
(**a**) Transfer characteristics, (**b**) field-effect mobility, and (**c**) output characteristics of InO*_x_* TFTs with various InO*_x_* thicknesses.

**Figure 5 nanomaterials-13-02568-f005:**
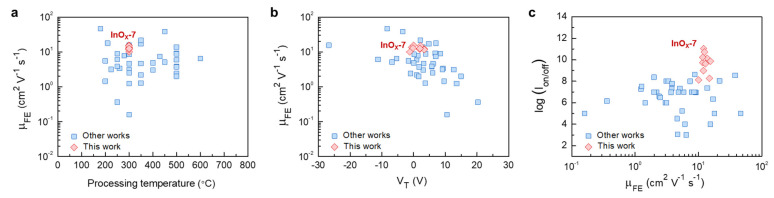
Benchmarks of (**a**) *μ*_FE_ versus processing temperature, (**b**) *μ*_FE_ versus *V*_T_, and (**c**) log(*I*_on/off_) versus *μ*_FE_. All works used SiO_2_ dielectric and solution-processed oxide semiconductors to create TFTs.

**Figure 6 nanomaterials-13-02568-f006:**
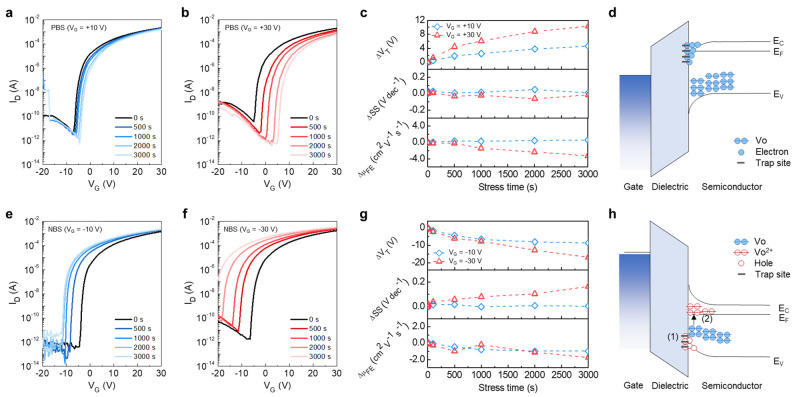
PBS with (**a**) *V*_G_ = 10 V and (**b**) *V*_G_ = 30 V. (**c**) Change in electrical parameters of InO*_x_* TFTs via positive bias stress. (**d**) PBS-induced degradation mechanism in InO*_x_* TFTs. NBS with (**e**) *V*_G_ = −10 V and (**f**) *V*_G_ = −30 V. (**g**) Change in electrical parameters of InO*_x_* TFTs via positive bias stress. (**h**) PBS-induced degradation mechanism in InO*_x_* TFTs.

## Data Availability

Not applicable.

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
