# Peer review of "Atomically Thin Amorphous Indium–Oxide Semiconductor Film Developed Using a Solution Process for High-Performance Oxide Transistors"

_nanomaterials, 2023, doi:10.3390/nano13182568_

Round 1

Reviewer 1 Report

I think that this one is a good paper and basically worth publishing in the journal.

The material name “InOx” had better be included in the title, because this is one of the important points of this work.

They fabricated device-patterns by direct UV irradiation of the spin-coated films. This seems a unique, interesting technique, and had better be described in more details. According to Fig.1, it seems that the irradiated parts remain, do not dissolve in the developing process. Then, does the UV irradiation affect the properties of the films and devices ? Have they tried different irradiation conditions?

One of the common problems in the spin coating is thickness uniformity. How is variation in thickness within a substrate ?

I am not certain that such high mobility can result from percolation conduction. They should compare them with literature values of mobility of percolation conduction in similar materials, and discuss whether the conduction is actually due to percolation. (In addition to the power law behavior, the absolute values need to be discussed.)

Author Response

We appreciate your thoughtful review. Please see the attachment.

Reviewer 2 Report

The paper presents thin film transistors with low temperature solution processed thin InOx channel. It is well organized and the results are correctly described but may be not fully interpreted. State-of-the-art performance is demonstrated by using an optimized 3.12 nm InOx channel.

Comments:
- Fig. 4a: In the transfer characteristics, Vd is fixed at 30V, but two curves can be seen. Does this correspond to hysteresis curves? In that case, the minimum values of gate current are different for InOx-5 and -7 with around 30V shift between the two minimum, and for InOx-9 and -11 where the shift is only around 10V. Is there some explanation for that?
- Fig 4a: What is the origin of large negative VT shift when molarity increases? Is it relative to oxygen vacancies in InOx?
- Line 211: why do the percolation effects in InOx decrease as the thickness of InOx increases?
 - Fig. 4c: What are the Vg values for Id-Vd curves?
- In Table S1, are the various data concerning the present work, relative to the variability of the process? Or they differ by other technological parameters?
-  In the conclusion, it is mentioned that best device performance is obtained for an optimized InOx thickness of 3.12nm. This value is the result of the solution processed material properties. From theoretical point of view, for an homogeneous material, what is the optimized InOx thickness and how this value is modified taking into account the variation of the properties with molarity?

Author Response

(The authors gave the same response as above.)

Reviewer 3 Report

The authors skillfully manipulated the solution concentration of InOx semiconductors, effectively tuning both bandgap and thickness, culminating in the creation of remarkably competent field-effect transistors (FETs). However, a careful review of the claims made in the manuscript reveals striking similarities between the current study and previous research, as well as previous endeavors by the same group of authors. Notable examples of such precedent literature include the following research articles below:

- Electronics 11 (2022) 2822 DOI: 10.3390/electronics11182822

- Chemical Engineering Journal 441 (2022) 135833 DOI: 10.1016/j.cej.2022.135833

- ACS Nano 9 (2015) 4, 4572-4582 DOI: 10.1021/acsnano.5b01211

- ACS Applied Materials & Interfaces 7 (2015) 782-790 DOI: 10.1021/am5072139

Research has been conducted on various sol-gel based oxide semiconductors, and their main claims include attributes such as the feasibility of thin film deposition, low processing temperatures, and fabrication on flexible substrate. In many respects, these claims are very similar or analogous to the claims made by the authors. Given this context, the claims and findings presented in the authors’ manuscript should ideally be supported by arguments that are different or novel from previous works.

Furthermore, a processing temperature of 300 degrees cannot be considered low, especially when compared to numerous previously reported cases. In addition, the attempts to fabricate thin films using highly dilute solutions did not result in any discernible improvement in power efficiency or statistically significant metrics of the fabricated devices. Addressing these discrepancies and introducing innovative aspects is imperative to align the research with the existing points of the reported literature.

Minor wrong typos, but English is not the matter.

Author Response

(The authors gave the same response as above.)

Round 2

Reviewer 2 Report

The authors clearly answered to the comments. The paper is publishable in the current version

Author Response

We greatly appreciate the reviewer's positive comments.

Reviewer 3 Report

Thank you for the author's comments. I have read the authors' comments carefully, and it is certainly impressive to see the performance of different electronic devices by controlling the thickness down to the atomic scale through a solution-process. 

However, since it is a "solution process" rather than an ALD process, I believe that the performance and yield of the device will be very irregular depending on the surface and morphological defects (e.g., pin-holes, defect voids, etc.) and quality of the thin film due to the "atomic scale" of thin-films. The authors also represented the isolated channel regions in the patterning process for many devices in a batch, but all characteristic curves showed only one single curve. In this regard, I think it would be more convincing to show statistical indicators (e.g., statistical histograms, contour mapping) of many device performances by batch. I'm confident that if you strengthen these points, you'll make a deeper impression on your readers. 

Author Response

(The authors gave the same response as above.)

Round 3

Reviewer 3 Report

Research on the fabrication of extremely thin semiconductor films is a recent issue these days, so many readers will read with great interest this paper. In my opinion, this revised and improved paper will be presented with less skepticism and more positively.